# Intermittent Fasting: Does It Affect Sports Performance? A Systematic Review

**DOI:** 10.3390/nu16010168

**Published:** 2024-01-04

**Authors:** Javier Conde-Pipó, Agustín Mora-Fernandez, Manuel Martinez-Bebia, Nuria Gimenez-Blasi, Alejandro Lopez-Moro, José Antonio Latorre, Antonio Almendros-Ruiz, Bernardo Requena, Miguel Mariscal-Arcas

**Affiliations:** 1Health Science and Nutrition Research (HSNR, CTS-1118), Department of Nutrition and Food Science, University of Granada, Campus of Cartuja s/n, 18071 Granada, Spain; javiercondepipo@gmail.com (J.C.-P.); agusmora@correo.ugr.es (A.M.-F.); alexlopez@ugr.es (A.L.-M.); aalmendros@correo.ugr.es (A.A.-R.); 2Department Food Technology, Nutrition and Food Science, Campus of Lorca, University of Murcia, 30100 Murcia, Spain; manuelmb@um.es (M.M.-B.); joseantonio.latorre@um.es (J.A.L.); 3Nutrition Area, Faculty of Health Sciences, Catholic University of Avila, 05005 Ávila, Spain; nuria.gimenez@ucavila.es; 4Research and Development Department, Football Science Institute, 18016 Granada, Spain; bernardorequena@fsi.training; 5Instituto de Investigación Biosanitaria (ibs.GRANADA), 18012 Granada, Spain

**Keywords:** sport nutrition, athletic performance, fasting strategies, exercise

## Abstract

Intermittent fasting is one of the most popular types of diet at the moment because it is an effective nutritional strategy in terms of weight loss. The main objective of this review is to analyze the effects that intermittent fasting has on sports performance. We analyzed physical capacities: aerobic capacity, anaerobic capacity, strength, and power, as well as their effect on body composition. For this, a bibliographic search was carried out in several databases where 25 research articles were analyzed to clarify these objectives. Inclusion criteria: dates between 2013 and present, free full texts, studies conducted in adult human athletes, English and/or Spanish languages, and if it has been considered that intermittent fasting is mainly linked to sports practice and that this obtains a result in terms of performance or physical capacities. This review was registered in PROSPERO with code ref. 407024, and an evaluation of the quality or risk of bias was performed. After this analysis, results were obtained regarding the improvement of body composition and the maintenance of muscle mass. An influence of intermittent fasting on sports performance and body composition is observed. It can be concluded that intermittent fasting provides benefits in terms of body composition without reducing physical performance, maintenance of lean mass, and improvements in maximum power. But despite this, it is necessary to carry out new studies focusing on the sports field since the samples have been very varied. Additionally, the difference in hours of intermittent fasting should be studied, especially in the case of overnight fasting.

## 1. Introduction

The role that nutrition plays today in the sports performance of athletes is clear. However, in recent years, different dietary patterns and protocols have emerged that have tried to amplify or reduce the adaptations derived from physical exercise to try to achieve an improvement in the athlete’s sports performance [1,2,3]. Within these strategies, intermittent fasting has been acquiring special renewed interest due to its supposed effects on health and improvement of body composition in patients with different pathologies of great predominance nowadays. Intermittent fasting is a popular type of dietary pattern based on timed periods of fasting. This dietary protocol focuses on timed fasting/eating periods with different time intervals [4,5].

Despite this recent popularity, currently, its effects on performance do not seem to be clarified. This has special importance in the physical and cognitive performance of those athletes who frequently perform this type of dietary pattern or temporary caloric restrictions, as happens with Islamic athletes during the religious practice of Ramadan [5,6,7]. Therefore, the main objective of the present review was to investigate whether intermittent fasting intervenes in performance in professional athletes, evaluating the effects of this dietary intervention on aerobic and anaerobic capacity, strength and power, and body composition such as fat mass, muscle mass, and weight, which could affect sports performance. The study will focus on research findings regarding intermittent fasting.

## 2. Materials and Methods

This work is a systematized literature review based on scientific evidence documenting the effects of intermittent fasting in athletes and/or people who perform physical activity and its benefits on sports performance. This review was carried out in 2022 and registered in PROSPERO with code ref. 407024, and an evaluation of the quality or risk of bias assessment was performed by J.C.-P using the blinded Cochrane risk of bias tool.

For the literature review, the following databases were used: MEDLINE through the PubMed search engine, Web of Science (WOS), and Scopus. The keywords used were: “intermittent fasting”, “sport”, “performance”, and “exercise” where 4402 articles were obtained using Boolean connectors “or” between the words in different languages and “and” between the different ones (“intermittent fasting” OR “intermittent fasting” AND “sport” OR “sport” AND “performance” OR “performance” AND “exercise” OR “exercise”). The inclusion criteria that have been established are the following: dates established between 2013 and the present, due to the importance that is beginning to be given to intermittent fasting studies from this year onwards; free full texts; studies conducted in adult human athletes; languages of the articles in English and/or Spanish; it has been considered that intermittent fasting is mainly linked to sports practice and that this obtains a result in terms of performance or physical capacities. The exclusion criteria of the articles are the following: dates established after 2013, texts not available, and languages of the articles being other than English and/or Spanish.

Therefore, when filtering articles from 2013 to today, focusing on humans and adults and limiting the languages to Spanish and English, 114 articles were found. After eliminating duplicates, we obtained 92 papers. Once the titles were read, 34 were selected and 58 were eliminated because the title was not related to the topic of the systematic review. After reading the abstracts and observing the bibliographies, 12 articles were eliminated, but 3 were added from the bibliographies obtained by complete readings. This whole procedure was followed according to PRISMA standards [8], by means of which a flow chart was made (Figure 1). No deviation from these standards was found.

## 3. Results

Within the bibliography of this review, we found that the majority highlighted that the subjects were athletes [9,10,11,12,13,14,15,16,17,18,19,20]. Another did not specify if they were athletes [21], but due to their intervention, it was included, while the rest indicated that the subjects were physically active [22,23,24,25,26].

Most of them showed time-restricted feeding (TRF) with a 16 h fasting window and 8 h feeding window [9,10,11,12,14,16,19,22,23,24,27]. Other studies analyzed 14/8 fasting during the Ramadan period [12,18,28], while another article analyzed overnight fasting [15].

In addition to this, it should be noted that two studies used more variables in their studies, such as the intake of supplementation in addition to the fasting follow-up, namely hydroxy methyl butyrate (HMB) and two types of whey protein: whey protein concentrate (WPC) and hydrolyzed whey protein (WPH) [24,29]. Another article sought to compare the difference between protein-loaded and carbohydrate-loaded fasting [20]. For the most part, a control group with no fasting and an experimental group performing fasting are identified [23].

Data are shown in Table 1, based on performance, specifically aerobic performance, eight articles were included that used different tests, among them: 20 min cycling test [29], 10 km test [11], repeated sprints test [15,26], treadmill test [16], and test at 45% of maximum power [9]. Regarding anaerobic performance, six articles used different tests to evaluate different parameters of the sample such as stress tests [9,11], Wingate test [22,26,29], submaximal exercise [20], repeated sprints test [26], and interval training [20]. Regarding muscular strength and power, eight included studies that evaluated muscular strength through maximal strength and endurance strength tests and power through peak power (PPO) and average power (W) [10,14,18,19,24,25,26,27]. Finally, regarding body composition and health, we included the 15 studies that evaluated any body composition variable such as body fat mass, lean mass, and anthropometric folds [9,10,11,14,16,17,18,19,21,23,24,25,28,30,31,32,33].

## 4. Discussion

To begin with, we must consider that there is a wide variety of studies chosen in the review in terms of the difference of sports in the sample, the differences in gender and age, and the different interventions.

Firstly, we can affirm that intermittent fasting (any type) metabolically affects the body composition [9,10,11,12,14,23,24,25,26,27]. With this, we can concretize that it intervenes positively in performance since a reduction in body weight would be considered beneficial [9,11,25].

Intermittent fasting could be considered an adequate nutritional strategy to reduce body fat percentage to an adequate number for the athlete (between 6 and 12% in men and between 12 and 18% in women) and maintain lean mass or muscle mass [1,2,7,28]. A study published in 2021 with 14 active women who combined TRF with high-intensity training (HIIT) compared to HIIT with a normal diet observed a significant decrease in fat in the fasting group. It should be noted that the food log showed a non-voluntary caloric restriction of 10–20% per week [25]. A study with 50 healthy subjects combining a fasting intervention together with the performance of physical activity [28] observed a decrease in body mass index and body fat percentage.

It should be clarified that the effectiveness of the intermittent fasting protocols included in this review is closely linked to the time of application and to the type of population. Fasting is more effective in the medium and long term than in the short term [22,25], which raises the importance of adherence to treatment with this protocol. In turn, we should consider that in some studies [14,16,23,24,25] caloric restriction is combined with the follow-up of intermittent fasting. This aspect highlights conclusions obtained from recent studies where no superiority of different dietary protocols is observed, including intermittent fasting, if an adequate caloric deficit and a correct adherence to the dietary plan are reached [34,35]. Therefore, the effect of caloric restriction is a very important element to consider in relation to the reduction of fat mass.

Despite this, other studies show a greater adherence to the diet in those subjects who followed a nutritional plan with intermittent fasting compared to a diet without restrictions [23]. This aspect could be interesting considering the important role of a higher adherence for the adequacy of long-term fat loss and the need for individualization of the same [36].

Considering health in conjunction with intermittent fasting, a review published in 2022 points out that fasting controls body weight, improves insulin sensitivity, reduces systemic inflation, and strengthens the immune system [12]. Thus, it can be linked to injury prevention and recovery. Several reviews mention the benefits of intermittent fasting in improving metabolic health and insulin sensitivity [30], as well as in regulating glucose and certain lipid metabolism [32]. Certain metabolic processes in adipose tissue are mediated by endogenous clocks in our body such as adiponectin, the levels of this hormone can be altered by changes in sleep/wake or feeding/fasting cycles [32]. The TRF type of fasting appears to be an effective strategy to improve the levels of this hormone. Thirty-four men habituated to resistance training [10] found an increase in adiponectin levels and a decrease in leptin levels. The same author in another study with 16 cyclists [9] observed a trend of higher increase in adiponectin levels in the TRF group (+33%) compared to (+8%) in the normal diet group. Low levels of adiponectin have been associated with obesity, oxidative stress [10], and insulin resistance, whereas higher levels are associated with improved insulin sensitivity in adipose tissue [32].

This improvement in the treatment of different diseases has also been contrasted by other trials [37,38]. However, recent reviews show the need for long-term studies evaluating this type of intervention versus isocaloric restriction in humans to know whether the results may be different from those observed after similar weight loss achieved through modest continuous energy restriction [39,40,41].

With respect to the levels of certain inflammatory markers, some studies hypothesize that intermittent fasting may activate cellular mechanisms that enhance immune function [4]. In a group of elite cyclists, interleukin (IL-6) appeared to decrease in the TRF group with an upward trend in the normal diet group [9]. Also, in this same study, he observed a decrease in the neutrophil-lymphocyte ratio (NLR) in both groups, but it was significant relative to baseline values only in the TRF group. NLR is a biomarker of systemic inflammation that correlates with blood C-reactive protein levels, which were found to be decreased during Ramadan fasting in two studies of football players [17].

Finally, in terms of performance, we found both positive and negative aspects. It seems that during Ramadan (14/10), negative effects appear in terms of performance, which may be due to other aspects such as rest and hydration. Greater negative effects can be observed in elite athletes, but in amateur athletes, there is not much difference [12,18,19]. On the other hand, in the TRF modality (16/8), there do not seem to be differences in the performance of physical capacities: aerobic [9,11,13,16,27,29], anaerobic [11,13,19,26,29], and strength and power [10,12,14,19,20,22,24,27,29]. The results of this review thus show that, in terms of capacities, intermittent fasting does not have a negative impact on sports performance and could be considered an adequate type of diet for sports practice.

Recent reviews have also highlighted these varied findings, stressing the need for more research in this regard given the wide heterogeneity of protocols and measurable variables [19,42].

Nevertheless, the possible effect of this type of intervention on the athlete’s body composition and, therefore, its possible link with the athlete’s performance, should be considered [27,29].

Finally, the possible limitations observed during the elaboration of the work are presented, and due to these, it is not possible to obtain accurate conclusions. The review has had little variety of interventions, which does not allow us to identify a specific action protocol. The samples, tests, measurements, objectives, and variables of the different studies have also been a limitation. Some of them did not count calories during the interventions and because of this we cannot obtain a reliable conclusion about the results obtained, or in the case of the variables, not all the studies measured the same ones, so we cannot draw reliable conclusions in their entirety. Many of the articles look for short-term effects when it has been concluded that long- and medium-term effects are more effective. Most of the studies conclude that a follow-up of future interventions is necessary to continue monitoring the effects of fasting, so there seems to be a lack of evidence in the studies. Also, the difference in hours of intermittent fasting should be studied, especially in the case of overnight fasting. The diversity of the studies analyzed must be taken into account, which may affect the generalizability of the conclusions.

## 5. Conclusions

After carrying out this systematic review of the studies and the literature on how intermittent fasting affects athletes and sports performance, the questions and objectives posed at the beginning of the study can be answered. Referring to the main objective, the scientific evidence indicates that intermittent fasting does not negatively affect sports performance and does affect the improvement of body composition. Therefore, it may be an appropriate strategy for sports or athletes seeking control of lean mass and a decrease in fat mass, which can have a positive transfer in terms of performance, since by reducing their weight and specifically their fat mass they will be able to perform better in certain situations in which power capacity is used.

The performance of the aerobic capacity would be improved since the improvement of the corporal composition can help to the improvement of the resistance; this fact ends in an improvement of the power being related directly with the corporal weight. As far as strength is concerned, it is not compromised by intermittent fasting and remains at the same level. Intermittent fasting would be related to an increase or improvement of health and the immune system; this is also related to performance as it improves the ability to prevent injuries and promote recovery. Further studies would be needed to be more certain about the effects of intermittent fasting on sports performance. The heterogeneous samples make it difficult to have a definitive conclusion, but the data are favorable, and it could be recommended to follow intermittent fasting without reducing performance in athletes.

The case of Ramadan is different: it can be considered as a type of intermittent fasting/TRF in which the fasting period varies between 12 and 18 h depending on the season and location, unlike the other types of TRF, the restriction on the intake of any type of liquid and the fasting time lasts from sunrise to sunset. This causes disturbance in the sleep/wake and rest/activity cycles. Due to changes in sleeping and eating hours, physiological and psychological disturbances can be caused. There is evidence showing that fasting during Ramadan affects a decrease in performance in the YO-YO test, which measures VO2 max [22].

## Figures and Tables

**Figure 1 nutrients-16-00168-f001:**
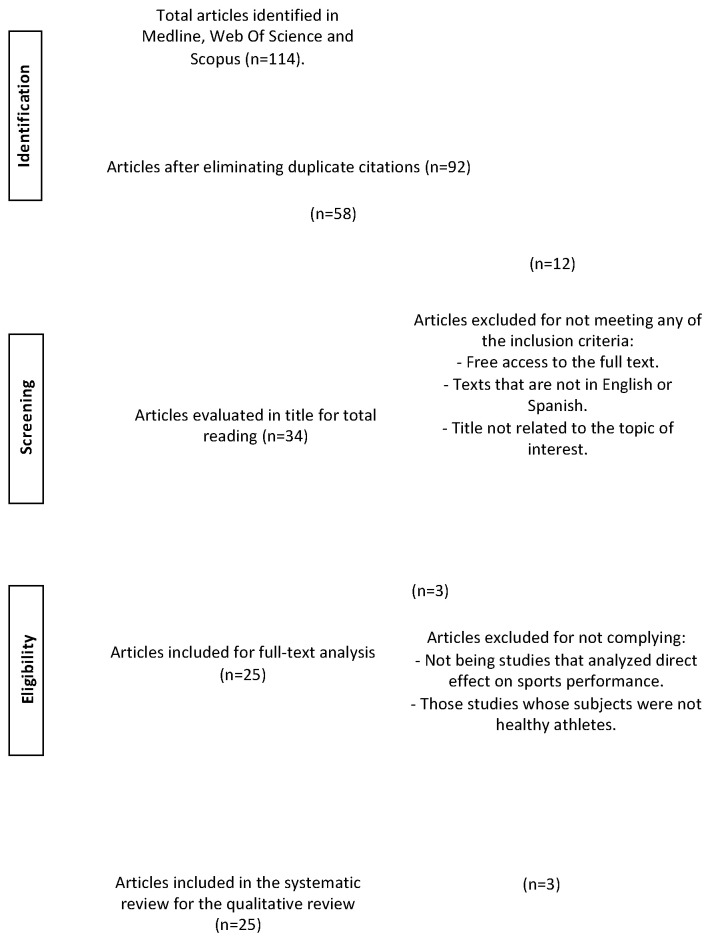
Flow chart for the selection of articles included in the systematic review.

**Table 1 nutrients-16-00168-t001:** Results of IF on aerobic capacity, anaerobic capacity, muscular strength and power, and body composition and health.

No.	Reference and Date	Impact Index	Type of Study	Study Size	Duration of Fasting	Objective of the Study	Parameters Analyzed	Conclusions
Aerobic capacity								
1.	Moro T, et al., 2020 [9]	5.159 (IF) 22/88 (Q1)	Experimental	16 young cyclists	Intermittent fasting TRF (16/8)	IF in 4 weeks of high-level resistance training.	Body composition, resting metabolism, and performance tests.	Does not affect performance.
2.	Kang J, et al., 2021 [27]	3.571 (IF) 57/109 (Q3)	Review	23 randomized studies	TRF fasting	Effects on metabolic and anthropometric parameters.	Strength, power, and aerobic capacity.	Does not reduce aerobic capacity.
3.	Tovar AP, et al., 2021 [11]	6.706 (IF) 15/90 (Q1)	Experimental	15 male runners	Intermittent fasting TRF 16/8	Effects on the performance of endurance runners.	Body composition, stress test, and 10 km test.	No effect on performance.
4.	Aird TP, et al., 2018 [13]	3.631 (IF) 11/83 (Q1)	Meta-analysis	46 studies	NT	To determine the effects of IF on aerobic and anaerobic exercise performance.	Aerobic capacity.	Aerobic exercise performance does not differ when following IF vs. other nutrition.
5.	Terada T, et al., 2019 [15]	1.432 (IF) 67/85 (Q4)	Experimental	20 participants	Overnight fasting	Effects on subjects in sprint training and aerobic capacity.	Aerobic capacity.	Improved sprint fasting vs. carbohydrate abundance.
6.	Brady AJ, et al., 2021 [16]	6.289 (IF) 9/88 (Q1)	Experimental	17 participants	Fasting TRF (16/8)	Effect of 8 weeks of TRF in conjunction with training.	Body composition, aerobic capacity, and biomarkers.	No alteration in endurance running performance indices.
7.	NaHarudin, et al., 2018 [26]	2.376 (IF) 29/83 (Q2)	Experimental	20 participants	Intermittent fasting	Effect of IF on high-intensity exercise, Wingate test, and HIIT cycling test.	Wingate test.	Attenuated performance at the start of practice.
8.	Aird TP, et al., 2021 [29]	5.900 (IF) 36/146 (Q1)	Experimental	28 male participants	Intermittent fasting	Compare performance and metabolic adaptations of short-term SIT with fasting and with WPH or WPC supplementation.	Body composition, aerobic exercise.	No significant results.
Anaerobic capacity								
1.	Moro T, et al., 2020 [9]	5.159 (IF) 22/88 (Q1)	Experimental	16 young cyclists	Intermittent fasting TRF (16/8)	IF in 4 weeks of high-level resistance training.	Body composition, resting metabolism, and performance test.	No effect on performance.
2.	Correia JM, et al., 2021 [22]	4.614 (IF) 100/279 (Q2)	Experimental	12 healthy males	Fasting TRF 16/8	Short- and long-term effects in trained young people.	Body composition and Wingate test.	No significant results in terms of performance improvement.
3.	Terada T, et al., 2019 [15]	1.432 (IF) 67/85 (Q4)	Experimental	20 participants	Overnight fasting	Effects on subjects in sprint training and aerobic capacity.	Aerobic capacity.	Improved sprint fasting vs. carbohydrate abundance.
4.	Naharudin, et al., 2018 [26]	2.376 (IF) 29/83 (Q2)	Experimental	20 participants	Intermittent fasting	Effect of IF in high-intensity exercise, Wingate test, and HIIT cycling test.	Wingate test, Body composition, aerobic exercise.	Attenuated performance at the start of practice.
5.	Aird TP, et al., 2021 [29]	5.900 (IF) 36/146 (Q1)	Experimental	28 male participants	Intermittent fasting	Compare performance and metabolic adaptations of short-term SIT with fasting and with WPH or WPC supplementation.	Aerobic and anaerobic performance.	No significant results.
6.	Rothschild JA, et al., 2021 [20]	6.706 (IF) 15/90 (Q1)	Experimental	17 trained cyclists and triathletes	Intermittent fasting	Effects versus a protein-rich and a carbohydrate-rich meal on cycling performance.	Submaximal exercise, high-intensity exercise.	No difference versus CHO in HIIT. Like PRO, uncompromised performance in shorter duration and higher intensity sessions.
Muscular strength and power								
1.	Moro T, et al., 2016 [10]	3.786 (IF) 30/128 (Q1)	Experimental	34 participants	TRF (16/8)	Effects during endurance training in healthy males.	Body composition, strength, and biomarkers.	Improvement of biomarkers related to health, fat loss, and maintenance of muscle mass.
2.	Kang J, et al., 2021 [27]	3.571 (IF) 57/109 (Q3)	Review	23 randomized studies	TRF fasting	Effects on metabolic and anthropometric parameters.	Strength, power, aerobic capacity.	Improvements in body composition and no alteration in muscle mass synthesis.
3.	Tinsley GM, et al., 2017 [14]	2.576 (IF) 22/81 (Q2)	Experimental	18 participants	TRF fasting	To examine changes in body composition and strength in strength training in males.	Strength and body composition.	Variation in fat mass loss versus diet, but not in muscle mass gain.
4.	Tinsley GM, et al., 2019 [24]	6.766 (IF) 6/89 (Q1)	Experimental	Healthy women aged 18–30 years	TRF fasting	TRF + HMB in strength training vs TRF without HMB.	Body composition and muscle performance.	TRF did not slow adaptations in hypertrophy and performance vs. other diets.
5.	Martínez-Rodríguez A, et al., 2021 [25]	4.614 (IF) 100/279 (Q2)	Experimental	14 active women	Intermittent fasting	Effects of HIIT training and muscular and anaerobic performance.	Body composition, grip strength, jumping, Wingate cycling test.	Decreased fat mass and increased jumping performance.
6.	Naharudin, et al., 2018 [26]	2.376 (IF) 29/83 (Q2)	Experimental	20 participants	Intermittent fasting	Effect of IF in high-intensity exercise, Wingate test, and HIIT cycling test.	Wingate test, Body composition, aerobic exercise.	Attenuated performance at the beginning of practice.
7.	Abaïdia AE, et al., 2020 [18]	11.140 (IF) 2/88 (Q1)	Meta-analysis	11 studies	Fasting 14/10 (Ramadan)	Effects of 1 month of Ramadan on physical performance.	Aerobic performance, maximal power, strength, jump height, sprints.	No decrease in performance if nutrition is correct.
8.	Correia JM, et al., 2020 [19]	5.719 (IF) 17/88 (Q1)	Experimental	Individuals between 18 and 39 years	Intermittent fasting	Effects on sports performance.	Muscular strength, aerobic capacity, anaerobic capacity, and body composition.	Positive results in fat mass reduction, without significant results in terms of strength.
Body composition and health								
1.	Moro T, et al., 2020 [9]	5.159 (IF) 22/88 (Q1)	Experimental	16 young cyclists	Intermittent fasting TRF (16/8)	IF in 4 weeks of high-level endurance training.	Body composition, resting metabolism, and performance testing.	Improved body composition and inflammatory markers.
2.	Moro T, et al., 2016 [10]	3.786 (IF) 30/128 (Q1)	Experimental	34 participants	TRF (16/8)	Effects during endurance training in healthy males.	Body composition, strength, and biomarkers.	Improved health, fat loss, and maintenance of muscle mass.
3.	Hosseini S, et al., 2015 [28]	NT	Experimental	50 healthy subjects	Ramadan	Effects of Ramadan and physical activity on biochemical parameters.	Body weight, fat percentage, biomarkers.	Reductions in anthropometric parameters, lower cholesterol.
4.	Laza V. 2020 [30]	NT	Magazine article	NT	TRF fasting	Effects on the performance and health of athletes.	Biomarkers, body composition.	Decreased blood glucose, body fat, cholesterol, testosterone levels, improved insulin sensitivity, increased hepcidin levels, improved immune system, and maintenance of muscle mass.
5.	Zouhal H, et al., 2020 [31]	NT	Review	71 studies	ICR, ADF, and TRF fasts	Identifying the effects of IF together with physical exercise.	Body composition, metabolic adaptations, sports performance.	Decreased circulating insulin levels and improved glucagon levels. Reduction of body fat.
6.	Tovar AP, et al., 2021 [11]	6.706 (IF) 15/90 (Q1)	Experimental	15 male runners	Intermittent fasting TRF 16/8	Effects on the performance of endurance runners.	Body composition, stress test, and 10 km test.	Improvements in fat mass reduction and muscle mass maintenance.
7.	Isenmann E, et al., 2021 [23]	6.706 (IF) 15/90 (Q1)	Experimental	35 subjects	TRF 16/8	Effects on body composition and adherence.	Weight, fat mass, BMI.	Improvements in weight, body composition, BMI, and hip and waist circumference.
8.	Haupt S, et al., 2021 [32]	6.064 (IF) 75/297 (Q2)	Review	NT	TRF 16/8	Summarize fasting information on metabolic and hormonal responses.		Improvements in blood pressure, insulin sensitivity, and body composition. Increased lipid utilization.
9.	El-Outa A, et al., 2022 [33]	0.678 (SJR) (Q2)	Experimental	80 participants	TRF 16/8	Assess VO2max in addition to other parameters.	VO2max, weight, body composition, biomarkers.	Reductions in glucose levels, LDL, HDL, and body weight. No significance in VO2max.
10.	Tinsley GM, et al., 2017 [14]	2.576 (IF) 22/81 (Q2)	Experimental	18 participants	Fasting TRF	Examine changes in body composition and strength in strength training in males.	Strength and body composition.	Variation in fat mass loss vs. diet, but not in muscle mass gain.
11.	Brady AJ, et al., 2021 [16]	6.289 (IF) 9/88 (Q1)	Experimental	17 participants	Fasting TRF (16/8)	Effect of 8 weeks of TRF together with training.	Body composition, aerobic capacity, and biomarkers.	Decrease in fat mass.
12.	Martínez-Rodríguez A, et al., 2021 [25]	4.614 (IF) 100/279 (Q2)	Experimental	14 active women	Intermittent fasting	Effect of HIIT training and muscular and anaerobic performance.	Body composition, gripper strength, jumping, Wingate cycling test.	Decrease in fat mass.
13.	Naharudin, et al., 2018 [26]	2.376 (IF) 29/83 (Q2)	Experimental	20 participants	Intermittent fasting	Effect of IF on high-intensity exercise, Wingate test, and HIIT cycling test.	Wingate test, body composition, aerobic exercise.	Attenuated performance at the beginning of practice.
14.	Hammouda O, et al., 2013 [17]	3.534 (IF) 8/55 (Q1)	Experimental	15 soccer players	Fasting 14/10 (Ramadan)	Effects of Ramadan on lipoprotein fluctuation during exercise.	Body composition, biomarkers.	Reductions in fat mass and LDL without affecting muscle mass and increase in HDL (significant reduction in YO-YO test).
15.	Correia JM, et al., 2020 [19]	5.719 (IF) 17/88 (Q1)	Experimental	Individuals between 18 and 39 years old	Intermittent fasting	Effects on sports performance.	Muscle strength, aerobic capacity, anaerobic capacity, and body composition.	Positive results in fat mass decrease, no significant results in strength.

## Data Availability

There are restrictions on the availability of data for this trial due to the signed consent agreements around data sharing, which only allow access to external researchers for studies following the project’s purposes. Requestors wishing to access the trial data used in this study can make a request to mariscal@ugr.es.

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
