# Peer review of "Intermittent Fasting: Does It Affect Sports Performance? A Systematic Review"

_nutrients, 2024, doi:10.3390/nu16010168_

Round 1

Reviewer 1 Report

Comments and Suggestions for Authors

Thank you for submitting the manuscript "Intermittent fasting: does it affect sports performance? A systematic review" to Nutrients. The review article has an interesting topic and appears to have been well conducted, including PROSPERO registration. However, I have some suggestions to improve the manuscript and have acceptable quality to be accepted in this important journal.

- Line#19: I think that in this case it would be better to correct the verb tense to past since this is a report of work that has already been carried out.

- Line#25: not just any type of fasting, but rather intermittent fasting which is a specific strategy.

- Keywords: avoid using keywords that are already in the title to ensure the largest search engine list for your manuscript.

- Line#32: redundant because it is clear that nutrition can cause benefit or harm to any human being. I believe it can be better explained to make it clearer.

- In general, the introduction needs to be improved to make it clearer, for example, what intermittent fasting is among other basic concepts that will be reported throughout the text.

- Line#57: why repeat the terms if they are the same?

- It is important to include when the research was carried out.

- Table 1, 2, 3 and 4: it would be interesting if the authors created a scale or a way to compare the works between them in terms of the conclusion obtained. My suggestion is based on the fact that minimum, maximum, medium, little and much demonstrate different escalations for each person reading the manuscript. Furthermore, these practically blank pages between the tables make it difficult to read the manuscript. Perhaps it would be better if the four tables have just one separating the results of each "part" with a line and a new title.

- Line#124: "improvement of body composition" doesn't seem like an appropriate expression to me. What is improvement in this case for the area of sports nutrition?

- LIne#131: check this quote and the others throughout the text as they do not seem to be within the standard of this journal.

- Line#130: weight or percentage?

- I missed in the discussion that the authors bring from the literature how intermittent review acts metabolically, for example by reducing body mass (line#130).

- The limitations need to be better explained and for example, the difference in hours of intermittent fasting, especially in the case of overnight, needs to be included.

Comments on the Quality of English Language

Minor editing of English language required.

Author Response

REVIEWER 1

Thank you for submitting the manuscript "Intermittent fasting: does it affect sports performance? A systematic review" to Nutrients. The review article has an interesting topic and appears to have been well conducted, including PROSPERO registration. However, I have some suggestions to improve the manuscript and have acceptable quality to be accepted in this important journal.

RESPONSE: The authors are very grateful to Reviewer 1, whose comments and corrections have substantially improved the manuscript.

- Line#19: I think that in this case it would be better to correct the verb tense to past since this is a report of work that has already been carried out.

RESPONSE: Corrected

- Line#25: not just any type of fasting, but rather intermittent fasting which is a specific strategy.

RESPONSE: Corrected

- Keywords: avoid using keywords that are already in the title to ensure the largest search engine list for your manuscript.

RESPONSE: Corrected

- Line#32: redundant because it is clear that nutrition can cause benefit or harm to any human being. I believe it can be better explained to make it clearer.

RESPONSE: Corrected

- In general, the introduction needs to be improved to make it clearer, for example, what intermittent fasting is among other basic concepts that will be reported throughout the text.

RESPONSE: Corrected

- Line#57: why repeat the terms if they are the same?

RESPONSE: Corrected

- It is important to include when the research was carried out.

RESPONSE: Included

- Table 1, 2, 3 and 4: it would be interesting if the authors created a scale or a way to compare the works between them in terms of the conclusion obtained. My suggestion is based on the fact that minimum, maximum, medium, little and much demonstrate different escalations for each person reading the manuscript. Furthermore, these practically blank pages between the tables make it difficult to read the manuscript. Perhaps it would be better if the four tables have just one separating the results of each "part" with a line and a new title.

RESPONSE: Corrected

- Line#124: "improvement of body composition" doesn't seem like an appropriate expression to me. What is improvement in this case for the area of sports nutrition?

RESPONSE: Corrected

- LIne#131: check this quote and the others throughout the text as they do not seem to be within the standard of this journal.

RESPONSE: Corrected

- Line#130: weight or percentage?

RESPONSE: Corrected

- I missed in the discussion that the authors bring from the literature how intermittent review acts metabolically, for example by reducing body mass (line#130).

RESPONSE: Corrected

- The limitations need to be better explained and for example, the difference in hours of intermittent fasting, especially in the case of overnight, needs to be included.

RESPONSE: Corrected

Reviewer 2 Report

Comments and Suggestions for Authors

Abstract:

·         While the abstract outlines the general goal and findings, it could benefit from more specific details about the methodology and the scope of the literature reviewed. For instance, mentioning the criteria for selecting the 25 research articles could enhance the reader's understanding of the study's rigor.

·         The results and conclusions are mentioned briefly. Expanding on these, particularly how intermittent fasting influences sports performance and body composition, would provide a more comprehensive understanding for the reader.

·         While the abstract mentions the need for further studies, a more explicit statement on the implications of the current findings for future research would be beneficial.

1. Introduction:

·         The introduction could benefit from a more in-depth discussion of previous research findings related to intermittent fasting and sports performance. This would provide a stronger foundation for understanding the study's significance.

·         Some parts of the introduction are a bit lengthy and could be more concise. Shortening and refining these sections would improve readability and focus.

·         While the introduction sets the stage for the study, it could more explicitly state the research objectives and any hypotheses the study intends to test. This would give readers a clearer understanding of the study's aims from the outset.

·         A brief preview or link to the methodology section explaining how the study will address the research gap could be beneficial, providing a seamless transition from the introduction to the rest of the paper.

2. Materials and Methods

·         While the section details the databases and keywords, it could benefit from a more explicit explanation of the search strategy, including date ranges for the searches.

·         The rationale behind specific exclusion criteria, such as the focus on studies from 2013 onwards, is not clearly explained. Elaborating on this would enhance the reader's understanding of the study's scope.

·         A more detailed description of the article selection and screening process, including the number of articles screened at each stage and reasons for exclusion, would provide a clearer picture of the review process.

·         The manuscript mentions an evaluation of quality or risk of bias but does not provide details on how this was conducted. A more thorough explanation would add to the methodological rigor.

·         While the manuscript adheres to PRISMA standards, including a flow chart, it would be beneficial to discuss any deviations from these standards, if applicable.

3. Results

·         While the section outlines various findings, it could benefit from a clearer and more detailed presentation of the results. Including specific data points or summary statistics would provide a more comprehensive understanding of the findings.

·         Ensuring uniformity in how results are reported across different studies would enhance the readability and comparability of the findings.

·         The section could better address the variability in results due to different fasting protocols, athlete types, and performance measures used in the studies.

·         Providing more context about how these findings fit into the existing body of knowledge on the topic would be beneficial. Comparing these results with previous studies or highlighting novel insights would enhance the section's value.

4. Discussion

·         The discussion could benefit from a more critical analysis of the diversity of the studies and how this impacts the generalizability of the findings.

·         While positive aspects of intermittent fasting are highlighted, the discussion could more thoroughly address any contradictory findings or negative aspects reported in the studies.

·         The relationship between improved body composition and actual sports performance could be more explicitly detailed.

·         The limitations of the reviewed studies are mentioned, but a more detailed discussion on how these limitations might affect the conclusions would strengthen this section.

·         While the need for more research is mentioned, specific recommendations for future studies, such as particular areas of focus or methodologies, would be beneficial.

Comments on the Quality of English Language

The English used in the manuscript is generally understandable and coherent. It does not present significant difficulties in comprehension. However, there are minor areas where the language could be polished for clarity and fluency. 

Author Response

REVIEWER 2

Abstract:

While the abstract outlines the general goal and findings, it could benefit from more specific details about the methodology and the scope of the literature reviewed. For instance, mentioning the criteria for selecting the 25 research articles could enhance the reader's understanding of the study's rigor.

The results and conclusions are mentioned briefly. Expanding on these, particularly how intermittent fasting influences sports performance and body composition, would provide a more comprehensive understanding for the reader.

While the abstract mentions the need for further studies, a more explicit statement on the implications of the current findings for future research would be beneficial.

RESPONSE: Corrected

  1. Introduction:

The introduction could benefit from a more in-depth discussion of previous research findings related to intermittent fasting and sports performance. This would provide a stronger foundation for understanding the study's significance.

Some parts of the introduction are a bit lengthy and could be more concise. Shortening and refining these sections would improve readability and focus.

While the introduction sets the stage for the study, it could more explicitly state the research objectives and any hypotheses the study intends to test. This would give readers a clearer understanding of the study's aims from the outset.

A brief preview or link to the methodology section explaining how the study will address the research gap could be beneficial, providing a seamless transition from the introduction to the rest of the paper.

RESPONSE: Corrected

  1. Materials and Methods

While the section details the databases and keywords, it could benefit from a more explicit explanation of the search strategy, including date ranges for the searches.

The rationale behind specific exclusion criteria, such as the focus on studies from 2013 onwards, is not clearly explained. Elaborating on this would enhance the reader's understanding of the study's scope.

A more detailed description of the article selection and screening process, including the number of articles screened at each stage and reasons for exclusion, would provide a clearer picture of the review process.

The manuscript mentions an evaluation of quality or risk of bias but does not provide details on how this was conducted. A more thorough explanation would add to the methodological rigor.

While the manuscript adheres to PRISMA standards, including a flow chart, it would be beneficial to discuss any deviations from these standards, if applicable.

RESPONSE: Included and corrected

  1. Results

While the section outlines various findings, it could benefit from a clearer and more detailed presentation of the results. Including specific data points or summary statistics would provide a more comprehensive understanding of the findings.

Ensuring uniformity in how results are reported across different studies would enhance the readability and comparability of the findings.

The section could better address the variability in results due to different fasting protocols, athlete types, and performance measures used in the studies.

Providing more context about how these findings fit into the existing body of knowledge on the topic would be beneficial. Comparing these results with previous studies or highlighting novel insights would enhance the section's value.

RESPONSE: The authors have tried to improve aspects of the results, giving greater order and understanding to the data provided, even modifying the format of the tables.

  1. Discussion

The discussion could benefit from a more critical analysis of the diversity of the studies and how this impacts the generalizability of the findings.

While positive aspects of intermittent fasting are highlighted, the discussion could more thoroughly address any contradictory findings or negative aspects reported in the studies.

The relationship between improved body composition and actual sports performance could be more explicitly detailed.

The limitations of the reviewed studies are mentioned, but a more detailed discussion on how these limitations might affect the conclusions would strengthen this section.

While the need for more research is mentioned, specific recommendations for future studies, such as particular areas of focus or methodologies, would be beneficial.

RESPONSE: The authors have attempted to respond to the points detailed by reviewer 2 by improving aspects of the discussion and the limitations of the study.
